# Fruit Detection and Counting in Apple Orchards Based on Improved Yolov7 and Multi-Object Tracking Methods

**DOI:** 10.3390/s23135903

**Published:** 2023-06-25

**Authors:** Jing Hu, Chuang Fan, Zhoupu Wang, Jinglin Ruan, Suyin Wu

**Affiliations:** School of Mathematics and Computer Science, Wuhan Polytechnic University, Wuhan 430024, China; jhu@whpu.edu.cn (J.H.); hammerzpw@gmail.com (Z.W.); louxingchen217@gmail.com (J.R.); suyinwu2021@163.com (S.W.)

**Keywords:** yolov7, object detect, multi-object tracing, self-attention

## Abstract

With the increasing popularity of online fruit sales, accurately predicting fruit yields has become crucial for optimizing logistics and storage strategies. However, existing manual vision-based systems and sensor methods have proven inadequate for solving the complex problem of fruit yield counting, as they struggle with issues such as crop overlap and variable lighting conditions. Recently CNN-based object detection models have emerged as a promising solution in the field of computer vision, but their effectiveness is limited in agricultural scenarios due to challenges such as occlusion and dissimilarity among the same fruits. To address this issue, we propose a novel variant model that combines the self-attentive mechanism of Vision Transform, a non-CNN network architecture, with Yolov7, a state-of-the-art object detection model. Our model utilizes two attention mechanisms, CBAM and CA, and is trained and tested on a dataset of apple images. In order to enable fruit counting across video frames in complex environments, we incorporate two multi-objective tracking methods based on Kalman filtering and motion trajectory prediction, namely SORT, and Cascade-SORT. Our results show that the Yolov7-CA model achieved a 91.3% mAP and 0.85 F1 score, representing a 4% improvement in mAP and 0.02 improvement in F1 score compared to using Yolov7 alone. Furthermore, three multi-object tracking methods demonstrated a significant improvement in MAE for inter-frame counting across all three test videos, with an 0.642 improvement over using yolov7 alone achieved using our multi-object tracking method. These findings suggest that our proposed model has the potential to improve fruit yield assessment methods and could have implications for decision-making in the fruit industry.

## 1. Introduction

The online sale of fresh fruits has emerged as a significant channel for consumers in recent years. However, the sale of such products is typically not immediate and often involves pre-sales or similar methods. Accurate knowledge of the annual production of fruit products is thus crucial for setting pre-sale quantities effectively. Furthermore, fruit count serves as a crucial benchmark for assessing the productivity and quality of fruit production, as well as the overall efficiency of agricultural practices [1]. These observations underscore the importance of developing robust methods for evaluating fruit yield in agricultural settings.

Fruit counting is typically categorized into two types: detection-based fruit counting and regression-based fruit counting. The former is generally regarded as superior in performance, according to recent research [2]. For detection-based fruit counting, accurate fruit detection and localization are key determinants of model performance.

In the past few decades, numerous innovative approaches have been proposed to address the challenges associated with fruit detection in agricultural settings. Due to the complex growing environments of fruits, initial efforts often involve extracting color, shape, or other dominant features of fruits [3], followed by various processes. For instance, a wavelet transform-based approach was proposed by [4] to normalize fruit images with complex backgrounds, thereby reducing the impact of illumination. In addition, to address the complexity of environments in which sensors may not be able to discriminate data accurately, some researchers have introduced additional manual observation factors or used the wind to blow up leaves to reduce occlusion [5,6]. Other researchers have focused on color features [7], local texture-based features [8], and support vector machines for fruit detection [9]. Although many fruit detection methods have been proposed and have achieved relatively favorable results under specific conditions, they still fall short in tackling challenging situations such as changes in lighting, complex backgrounds, and variations in weather.

In recent years, deep learning has shown remarkable progress in various fields, particularly in computer vision tasks. The success of deep learning in image classification tasks, such as VGG [10] and ResNet [11], has led to the development of numerous methods for object detection. Object detection models can be broadly classified into two categories, one-stage models and two-stage models. While two-stage models, including Fast R-CNN [12], Faster R-CNN [13], and Sparse R-CNN [14], show better accuracy and recall due to decoupling localization and classification, one-stage models, such as the Yolo series [15,16,17] and Single Shot Multi-Box Detector [18], exhibit faster inference speed. Some researchers have introduced deep learning to fruit detection and counting; for example, [19] used the Yolo detection model to perform real-time rank classification of RBG images of olives, while [20] used Faster R-CNN and SSD to track and count different fruits separately.

On the other hand, as a downstream task of object detection, multi-object tracking has likewise seen significant advancements due to the continuous development of deep learning in recent years. Various multi-object tracking methods have been proposed based on object detection techniques, leveraging advances in convolutional neural networks (CNNs) and motion trajectory prediction. One such method is the simple online and real-time tracking (SORT) algorithm [21], which employs IOU intersection and ratio and Hungarian matching algorithms [22] to associate objects between different frames in real time. Another algorithm is Deep-SORT [23], which improves the appearance feature extraction capability of SORT by incorporating CNNs as appearance feature extractors. However, the overall detection speed of these models is limited by the complexity of the detection models used. With these in mind, [24] proposed a method to combine appearance features and detection for co-training, which significantly reduces the time consumption, but is difficult in terms of labeling and data collection as it requires the use of video data as training data.

In this study, we focus on the detection and counting of apples, one of the most commonly consumed fruits. To achieve real-time performance, we propose a detection framework that builds upon the current state-of-the-art Yolov7 [25] model by incorporating an attention mechanism to enhance object detection capabilities. In addition, we combine multi-object tracking methods and compare their performance to enable accurate inter-frame fruit counting. The results are presented in Section 3. The work in this study can be attributed to:
An improved Yolov7 architecture incorporating a self-attentive mechanism is proposed to enhance the detection performance for fruits.A cascaded multi-object tracking method combined with SURF is proposed to complete the detection and counting of apples inter-frames.


## 2. Materials and Methods

In this section, we will describe in detail the construction of the dataset, the model construction process, and the parameter assignment.

### 2.1. Building the Data Set

In this study, all apple data are obtained from publicly available datasets or free images on the Internet, among which we thank the authors of [26] for providing different quality classification apple datasets and [2] for the publicly available tropical fruit dataset, we segment the apple dataset from the tropical fruit dataset and subsequently take out the good apple from the dataset in [26] together as the apple for this experiment training data. Considering the agricultural scenario, the probability of bad apples in the harvest season is low, so bad apples are discarded. For each apple image, it can be an apple taken outdoors or placed indoors so that hidden information under different lighting may be obtained. In addition, different apples are allowed to block each other or be blocked by other objects, and the images at the boundary of the images are allowed to be truncated. Finally, we obtained 4246 apple images and divided the dataset into an 80% training set and a 20% validation set.

The unlabeled dataset provided in reference [26] poses a challenge for apple annotation, necessitating one of two options: either employ LabelImg [27] for image annotation on all unlabeled data or utilize an alternative approach for the already labeled tropical apple data. To this end, the modified yolov7 model was first trained on labeled data and subsequently deployed for inference on all unlabeled images, with the results duly recorded. The modified yolov7 model exhibits strong inference accuracy, especially in detecting and classifying images with simple backgrounds. As such, LabelImg is only needed for the secondary labeling of images with complex backgrounds. All labels generated are saved as txt files and sample images of the dataset are shown in Figure 1.

### 2.2. Data Augmentation

Data augmentation can be categorized into two distinct groups: pixel-level and spatial-level. In the former category, while the image may undergo some changes, the bounding box generally remains unaffected. Common pixel-level transformations include blurring, brightness and exposure adjustments, noise addition, Cutout, and Cutmix. On the other hand, spatial-level transformations have a broader scope of enhancement, not only modifying the image itself but also altering its size and orientation. These modifications can lead to the creation of more robust models.

This study employs a variety of data augmentation techniques, including random flip, random brightness adjustment, MixUP [28], and Mosaic [29], among others. MixUP involves the multiplication of two images with different factor ratios, and the resultant superimposed ratios are utilized to adjust the label file. On the other hand, Mosaic stitches together four images, enabling the model to detect objects outside the typical context while reducing the need for a large mini-batch size. Image resizing to 640 × 640 is conducted, while the aspect ratio of the fruit images remains unchanged, accomplished by dynamically filling the gray blocks. Example images are depicted in Figure 2.

### 2.3. The Improved YOLOv7 Counting Model

This section describes in detail the various methods and modules for improving Yolov7.

#### 2.3.1. Attention Mechanisms

The attention mechanism, originally introduced in [30], is a fundamental technique in Vision Transform [31] that enables the evaluation of correlations between different elements. By allowing neural networks to focus on critical regions of feature representations, the attention mechanism enhances the representation of essential features while suppressing non-important information. When analyzing images of fruit with complex backgrounds, implementing the attention mechanism effectively suppresses the representation of the background and improves focus on the fruit itself, leading to more accurate identification and location of fruit regions. Consequently, the overall performance of the model is enhanced. These findings demonstrate the utility of attention mechanisms for improving image analysis and highlight the importance of considering the benefits of this technique in future research endeavors.

Attention mechanisms are an essential component of modern neural network architectures, with applications ranging from image analysis to natural language processing. Based on their action level, attention mechanisms can be classified into three categories: channel attention mechanisms, spatial attention mechanisms, and hybrid attention mechanisms. Channel attention mechanisms focus on the importance of different channels and assign weight coefficients accordingly, as demonstrated in the SE (squeeze-and-excitation block) proposed by [32]. Spatial attention mechanisms, on the other hand, focus on different levels of importance in spatial information, as exemplified by the SA (spatial attention) proposed by [33]. Hybrid attention mechanisms take into account both spatial and channel interactions and perform joint information observation to select the most important part of the whole. Examples of such mechanisms include the CBAM (convolutional block attention module) proposed by [34] and the CA (coordinate attention) proposed by [35].

While both channel attention mechanisms and spatial attention mechanisms encapsulate only one level and are relatively less effective compared to hybrid attention mechanisms, we focus on the CBAM and CA mechanisms in this paper due to their superior performance. 

The CBAM attention mechanism focuses on the connection between channel attention and spatial attention, and its structure is illustrated in the figure below. CBAM employs an SE-like structure initially to obtain different channel weights, compressing all feature maps subsequently to obtain the spatial attention scores. Finally, the two scores are linearly multiplied to achieve the final output feature maps. The structure of the CBAM mechanism is illustrated in Figure 3.

The CA attention mechanism is distinct from the CBAM mechanism in that it encodes channel relationships and long-term dependencies, utilizing precise location information to do so. The structure of the CA mechanism is illustrated in Figure 4. By pooling and stitching feature layers in different directions, precise position information can be obtained, resulting in a more accurate determination of the region of interest. The final feature map information is obtained by capturing channel connections through 1 × 1 convolution and combining a residual unit.

#### 2.3.2. Yolo (You Only Look Once)

The field of computer vision has witnessed significant progress with the advancement of deep learning, leading to remarkable results in various research directions, including image classification, object detection, object segmentation, and key point detection. In particular, deep learning-based detection offers a powerful solution for fruit detection, as demonstrated by the successful application of the Yolo series. The Yolo series is widely recognized for its real-time detection performance, attributable to the single detection mechanism and powerful convolutional neural networks.

In this study, we choose the yolov7 model to construct the detection model. Unlike its traditional counterparts, Yolov7 supports single-GPU training, thereby allowing for seamless migration for training and deployment to edge GPUs. Yolov7 introduces parameter redistribution and dynamic label assignment; with these improvements, Yolov7 has achieved a balance of speed and accuracy, making it suitable for detecting most simple tasks using only the basic model. 

However, more accurate detection results are required in complex agricultural scenes with changes in lighting environment and a large number of occluded scenes, which suggests the need for further improvements to the Yolov7 model.

Therefore, in order to make the Yolov7 model obtain better results to enhance our multi-object tracking results, we combine the base Yolov7 with the two attention mechanisms mentioned in Section 2.3.1. The refined architecture of Yolov7 is demonstrated in Figure 5.

### 2.4. Cascade Multi-Object Tracking

Considering the above considerations, as the detection framework introduces a deep learning mechanism that can already identify the fruit in each frame well, the model needs to strike a better balance between speed and accuracy for edge GPU deployment, so in terms of tracking, this study proposes a cascading multi-object tracking method using SURF appearance feature extraction method to achieve a good balance between speed and accuracy, Additionally, Cascade-SORT, a method proposed by [36], also uses traditional machine learning with the VLAD method instead of deep convolutional networks, this method is similar to the one used in this study, but Cascade-SORT only uses a simple Yolov3 network, which does not allow it to achieve good results in terms of accuracy in the acquisition of detection frames, thus limiting the performance of the final results, and our model is based on the improved yolov7, which gives better detection results.

The comprehensive methodology of the object tracking technology employed in this research is presented in Figure 6. 

Details about the proposed methodology of this study are described as follows.

#### 2.4.1. Appearance Feature Extraction

Unlike Deep-SORT, which is geared towards pedestrian tracking detection and uses a deep learning network for feature extraction in appearance feature extraction, this significantly increases the computational speed of the model, whereas in an agricultural environment where the appearance of the same fruit changes very little during brief inter-frame movements, good results can still be achieved using traditional image feature extraction algorithms, while also being significantly faster than using convolutional neural networks.

Therefore, the use of SURF becomes a good choice. SURF constructs the scale space and Hessian matrix to complete the detection of scale invariant and rotation invariant feature points while using Haar wavelet response for the detection of feature descriptors, which can greatly improve the speed of feature extraction compared to other methods while ensuring accuracy. For each frame to be detected, the final appearance descriptor is a matrix of N × 128, where n is the number of feature points, as shown in Figure 7.

#### 2.4.2. Status Forecast

In order to resolve the occlusion of fruit by leaves, we use the Kalman filter [37] for the prediction of the current object trajectory. The state vector $\vec{x}$ of each object can be defined in eight dimensions as ${\left(x_{c},y_{c},a,h,v_{x},v_{y},v_{a},v_{h}\right)}^{T}$, the first four values represent the centroid coordinates, aspect ratio, and height of the object, and the last four values represent the velocity of movement of the first four values. Since the actual motion of the apple does not change drastically during the video shoot, we can treat the motion of the apple as a short-distance uniform motion and obtain the Kalman filter prediction as follows:(1)$${\hat{x}}_{t}={\left(x+v_{x},y+v_{y},a+v_{a},h+v_{h},v_{x},v_{y},v_{a},v_{h}\right)}^{T}$$

In addition, after the first prediction is completed, in order to make the prediction results more accurate, we use the match detection frame to correct the first Kalman filter of the corresponding trajectory accordingly after the object trajectory has been matched. As the initial velocity of the matching detection frame is 0, it can actually be regarded as a four-dimensional vector in the observation space, so we first map the first prediction result in the observation space and then find the observation margin by taking the difference between the matching detection frame and the prediction result, and finally obtain the motion-corrected result as follows:(2)$$x_{t}={\hat{x}}_{t}+{K\tilde{z}}_{}$$
where *K* is the Kalman gain in the observation space and $\tilde{z}$ is the observational margin that we derive in the observation space.

#### 2.4.3. Cascade Matching

For the frames to be detected obtained by the object detector, we first use a Kalman filter to initialize the state assignment, with each frame to be detected considered a unique ID; then, we use the SURF algorithm to extract the appearance features for each frame, repeating this operation, starting from the second frame, we will cascade the detected frames of the current frame to match the trajectories of the previous frame. The overall pipeline is shown in Figure 8.

For the motion characteristics between the two, we use the Mahalanobis distance for matching, and the Mahalanobis distance between the trajectory and the detection frame can be expressed as:(3)$$D_{m}=\sqrt{{\left(x_{t}-\;x_{d}\right)}^{T}S\left(x_{t}-x_{d}\right)}$$
where $x_{t}$ and $x_{d}$ denote the feature vectors of the trajectory and detection frame, respectively. In this study, the value is predicted by the Kalman filter, and *S* is the inverse of the Mahalanobis distance covariance matrix. The result of the Mahalanobis distance calculation is a real value that represents the similarity of the feature vectors between the trajectory and the detection frame, where a smaller value means that the two vectors are more similar to each other, i.e., closer in the feature space.

For appearance similarity detection, we use the cosine distance as a metric. Since the appearance features we extract using SURF are an *N* × 128-dimensional matrix, we assume that the feature matrices of the trajectory and detection frame are $F_{A}$ and $F_{B}$, respectively, the appearance similarity metric can be expressed as:(4)$$S_{featrure}=\frac{\sum_{i=1}^{N}\sum_{j=1}^{128}F_{A_{i,j}}\cdot F_{B_{i,j}}}{\sqrt{\sum_{i=1}^{N}\sum_{j=1}^{128}F^{2}{}_{A_{i,j}}}\cdot \sqrt{\sum_{i=1}^{N}\sum_{j=1}^{128}F^{2}{}_{B_{i,j}}}}$$

The above-mentioned motion and appearance features solve the matching problem in two separate dimensions. The motion similarity provided by the Mahalanobis distance combined with the Kalman filter prediction can solve the short-term matching problem well, while the appearance features provided by SURF can give a satisfactory result when faced with the longer-term occlusion problem.

We finally obtained three states of matching results: unmatched tracks, unmatched detections, and matched tracks. For matched tracks, we used the second Kalman filter to update the state information, including covariance matrix and velocity, according to their corresponding detection frames, and for unmatched tracks, in agricultural environments, there are cases where the fruit is obscured by fruit leaves or the fruit surface picture changes causing inaccurate matching. 

Therefore, we additionally invoke IOU threshold detection to make the determination and mitigate the effects caused by partial occlusion or fruit surface pictures, and for tracks that still fail to match, we set a life-cycle counter of 5 frames for unmatched tracks, considering that they are completely occluded by fruit leaves. If it is matched in the next five frames (including IOU matches), we reactivate it as a matched track, with the life cycle counter set to 0 for tracks in the deterministic state, and we discard unmatched tracks with a life cycle greater than five frames, treating them as completing one track. For unmatched successful detections, we treat them as a new track and use the Kalman filter for initialized state assignment and subsequent prediction.

### 2.5. Experimental Details

During the training phase of this experiment, AutoAnchor is initially used to determine the optimal anchor for the images in the training set. Subsequently, the image size is reduced to 640 × 640 pixels to facilitate training. YOLOv7 pre-trained weight files are utilized from the MS COCO dataset for transfer learning. To ensure accurate comparisons, three sets of control experiments are conducted with varying architectures, and their initial hyperparameters are set as follows: train-epochs is 300, batch-size is 32, the initial learning rate is 0.01, followed by cosine annealing scheduler with a final learning rate of 0.0001, and a warm-up of 20 epochs. Additionally, the training momentum and warm-up momentum are set to 0.937 and 0.8, respectively, and the weight decay is set to 0.0005. The box loss gain and cls loss gain are set to 0.05 and 0.5, respectively, while the image scale is set to 0.5, and the mosaic probability is set to 1.0. Mixup probability is set to 0.1, SGD is used as the training algorithm, BCELOSS is the loss function, and leakyReLU is the activation function, then we set sort_max_age to 5, sort_min_age to 3, sort_iou_thresh to 0.7, and hessianThreshold to 400. All experiments were conducted using a T4 GPU equipped with 16 GB memory for training purposes. It should be noted that the experimental environment was set up on Google Colab.

## 3. Results

### 3.1. Fruit Detection Results

To evaluate the performance of the model, it is necessary to calculate several indicators, including true positive (TP), false positive (FP), and false negative (FN), which are based on the intersection-over-union (IOU) value. Specifically, when the IOU of a predicted result is greater than 0.5 before the actual label, it is considered a TP detection; otherwise, it is considered an FP detection. In the absence of any detection or when the confidence level of the detected result is too low, it is considered an FN. Based on these parameters, we can calculate several performance metrics, including Precision, Recall, F1 score, and average precision (AP). Precision measures the model’s ability to correctly classify objects when detected, while Recall measures whether the model can accurately detect all objects. The F1 score provides an overall summary of the model’s performance. The AP is the area under the precision-recall curve, which is obtained by interpolating recall points and accuracy under different confidence levels. The mean average precision (mAP) is the AP index averaged over all categories. Although both AP and F1 can indicate the overall performance of the model, the AP index takes into account the confidence level effect. The calculation of all indicators is shown in Table 1.

According to the calculation method described in Table 1, we conducted ablation experiments on three different versions of the YOLOv7 model: YOLOv7, YOLOv7 with CBAM, and YOLOv7 with CA attention mechanisms, respectively. The results obtained are shown in Table 2, and as a reference object, we selected the experimental results of Yolov3 used in [36] for comparative observation, which are also given in Table 2.

Then we plot the loss curves for three different Yolov7 architectures in Figure 9.

### 3.2. Fruit Counting Results

To evaluate the effectiveness of the multi-object tracking methods, we consider that neither SORT, Cascade-SORT, nor the method proposed in this study involves deep learning when it comes to multi-object tracking and, therefore, can be employed as the same benchmark.

Additionally, the trained yolov7 model is directly utilized for video detection to assess the efficacy of multi-object tracking. Three fruit videos are selected for tracking and counting experiments, and the accuracy of counting is compared with the manual counting results using the following formula:(5)$$MAE=\frac{1}{m}\sum_{i=1}^{m}\left|\frac{{y_{test}^{\left(i\right)}-\hat{y}}_{test}^{\left(i\right)}}{{\hat{y}}_{test}^{\left(i\right)}}\right|$$

Among them, ${\hat{y}}_{test}^{\left(i\right)}$ represents the total number of manual counts in the acquired video sequence, and $y_{test}^{\left(i\right)}$ represents the counting results obtained by using two multi-objective tracking algorithms, where m is the number of videos to be detected, and i is the current first video. The selection of this indicator allows for visual observation of the overall nature of the model.

The comparative analysis of different counting methods is presented in Table 3.

## 4. Discussion

With the above results, we found that Yolov7 with CBAM showed a slight improvement in precision but a small decrease in Recall and a minor enhancement in overall performance. However, after incorporating the CA attention mechanism, the overall model demonstrated a 0.6% improvement in precision, a partial improvement in Recall, and a 0.4% improvement in mAP. These results suggest that the addition of attention mechanisms has a positive impact on the detection ability of apples using Yolov7. It is noteworthy that the model’s size did not show a significant effect on the running speed after incorporating the attention mechanisms.

Meanwhile, Yolov7 is overall better than the Yolov3 network. Considering the presence of migration learning, deep neural networks can enhance the overall generalization ability by combining fruit graphs with different feature distributions. Thus, the prediction model can be constructed with different parameters obtained by local migration learning. 

For the practical deployment in agricultural scenarios, due to the TBD approach architecture, the speed of the detector affects the overall tracking speed, so in the subsequent practical deployment, we may need to consider a pruning operation for the model, such as introducing the model compression method proposed by [38].

In this study, we found that the addition of the attention mechanism did not significantly affect the size scale of the model. We speculate that it may be because the added layer of the attention mechanism mainly deepens the model’s ability to understand small objects and does not add much complexity to the computation overall, and thus the size of the model is not significantly increased, which may make the deployment of the model easier because it will allow the compression of the model to focus mainly on the pruning optimization of the backbone network rather than the compression of the overall structure.

The counting results show that our proposed method has a larger improvement compared to SORT and Yolov7, with an improvement of 0.222 and 0.642 in the overall MAE, respectively, which may be due to the fact that SORT is too simple for image processing and does not take into account some complexities in the agricultural environment, while the simple use of Yolov7 makes it difficult to avoid too many repeated counts. As for Cacade-SORT, since a large part of the final overall performance of the multiple objects is affected by the performance of the detector, the overall result is improved by 0.084 MAE, thanks to the more powerful detection effect of our model.

However, it can be found from the results that the performance of the model cannot be stabilized very effectively for different video sequences, which may be due to the different recognition difficulty in different lighting and shading situations, and thus may lead to double counting or missed detection situations, This requires us to introduce additional metrics to further determine the performance of the model, such as ID switch, which can effectively determine whether the model is recounting, and MOTA, a composite performance metric, so in the future, we consider optimizing our metrics and constructing a fruit-related video dataset according to the method proposed by [24] to better solve the problem.

Summarizing the above discussion, we obtained the best experimental results using the Yolov7-CA detection head combined with SURF cascade matching, the relevant parameters of which are given in Section 2.4. These results indicate that the use of an attention mechanism can improve the performance of the detector, while the use of SURF to process images can also improve the effectiveness in multi-object tracking, which gives an improved idea for future research on fruit counting. Another thing to note is that the overall performance of the current fruit counting model is still mainly limited by the complexity of the fruit’s own growth environment, in addition to the problem of edge GPU deployment in real farms. Therefore, the overall size of the model and the detection speed need more consideration. In the future, we will combine the attention mechanism with improving the performance of different lightweight models to achieve a balance between speed and performance on fruit counting.

## 5. Conclusions

In this paper, we introduce a yolov7 structure with an attention mechanism to detect and count apple fruits by using a cascaded multi-object tracking technique with SURF extraction of appearance descriptions.

Our findings suggest that the inclusion of the attention mechanism enhances the model’s ability to detect apple fruits, while the multi-objective tracking technology improves the final counting outcomes. It is important to note that the experiments were conducted under certain experimental conditions, limiting the scope of our results. Specifically, we examined only two groups of attention mechanisms and found that the CA attention mechanism yielded superior results compared to the CBAM attention mechanism. However, further studies will be required to investigate the performance of additional attention mechanisms. In addition, we will introduce more judging criteria and use more multi-object tracking models in future research to find prediction models with a better balance between accuracy and speed and consider introducing model compression and pruning techniques in the future, which will be more conducive to practical edge GPU landing deployment with little impact on accuracy.

## Figures and Tables

**Figure 1 sensors-23-05903-f001:**
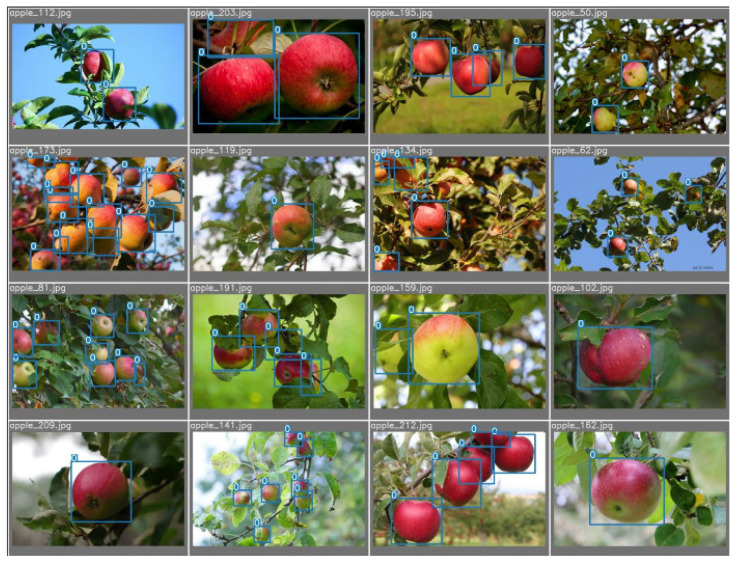
Partial apple GT image under natural lighting conditions.

**Figure 2 sensors-23-05903-f002:**
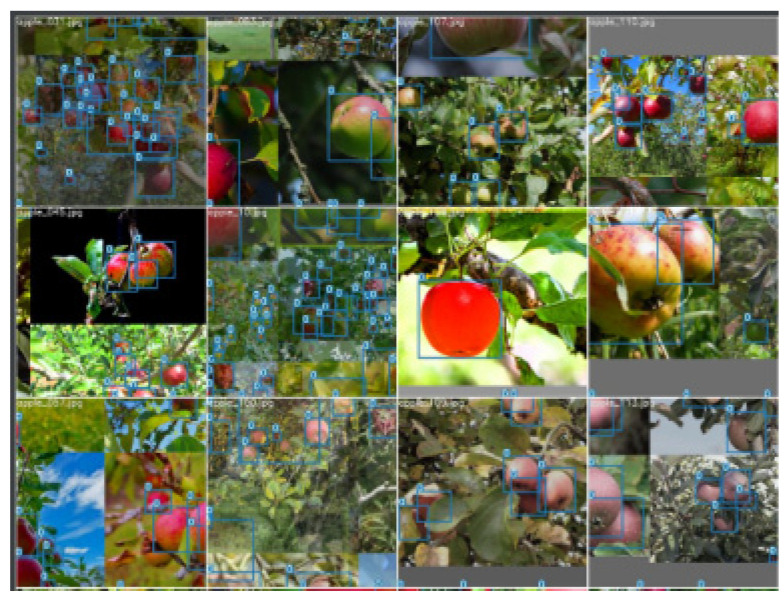
Partial apple image in natural environment after data enhancement.

**Figure 3 sensors-23-05903-f003:**
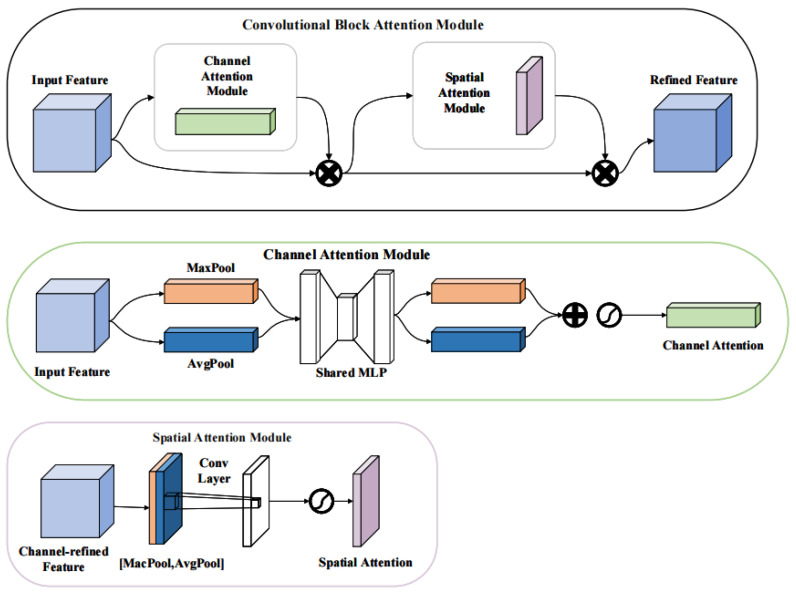
CBAM structure: The input features are augmented with additional information from linear cumulative channel attention and spatial attention.

**Figure 4 sensors-23-05903-f004:**
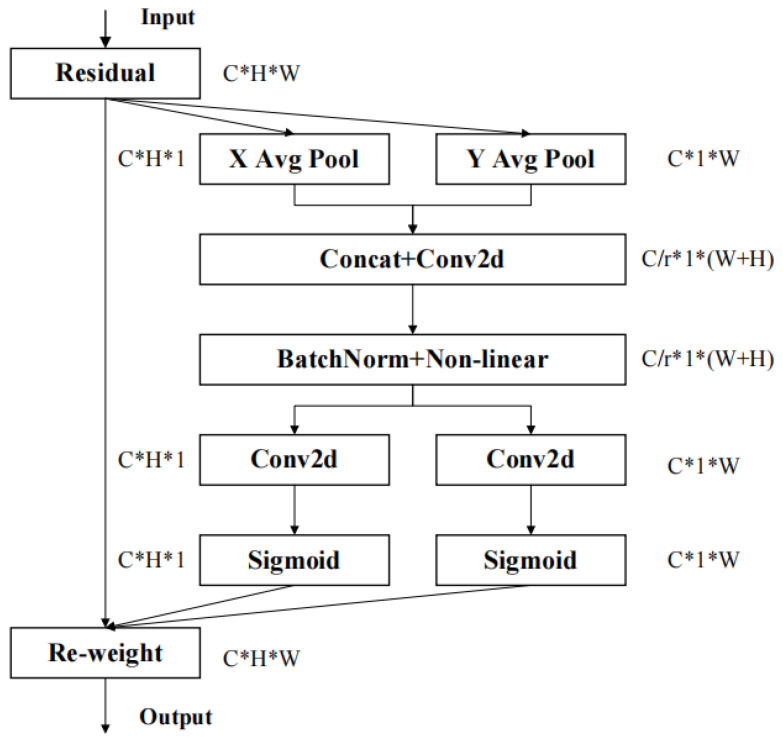
CA structure: Additional information units are obtained by finding regions of interest on the feature map, which are reparametrized and combined with the original feature residuals to form a new feature map.

**Figure 5 sensors-23-05903-f005:**
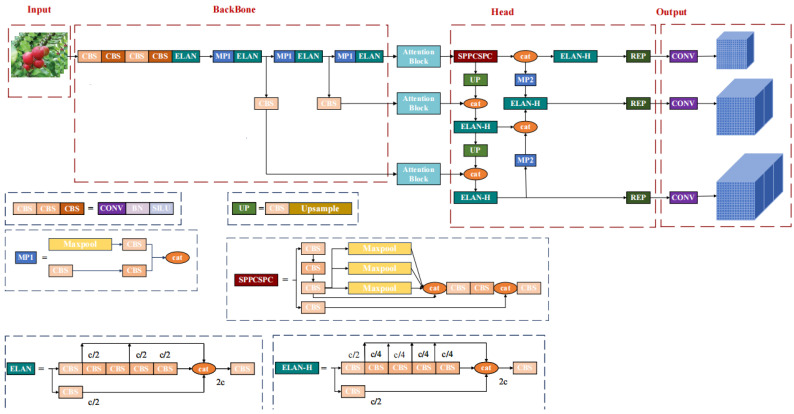
Improved yolov7 architecture: the images are resized to 640 × 640 and fed into the backbone network, additional attention mechanisms are added between the backbone network and the Head network for additional feature acquisition, and the improved yolov7 generates three different scales of predicted values to represent the classification and localization information of the detected objects.

**Figure 6 sensors-23-05903-f006:**
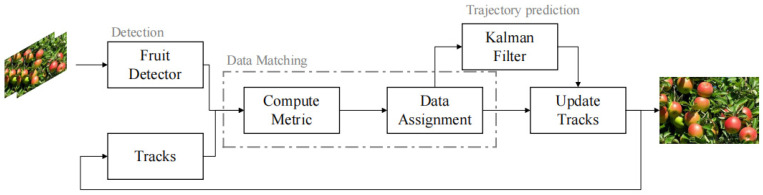
Fruit inter-frame tracking counting method pipeline.

**Figure 7 sensors-23-05903-f007:**
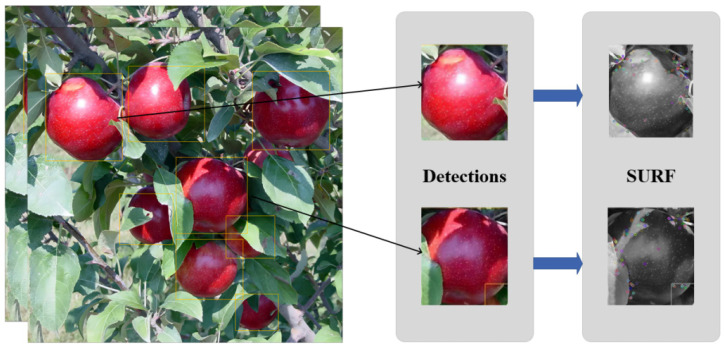
Apple feature descriptors extracted using SURF.

**Figure 8 sensors-23-05903-f008:**
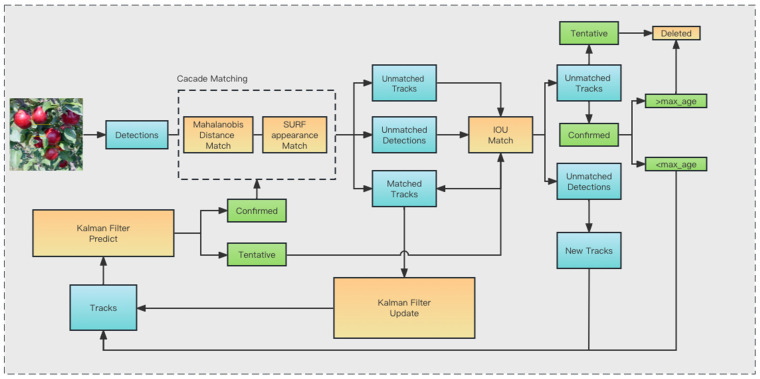
Multiple Objective tracking pipeline.

**Figure 9 sensors-23-05903-f009:**
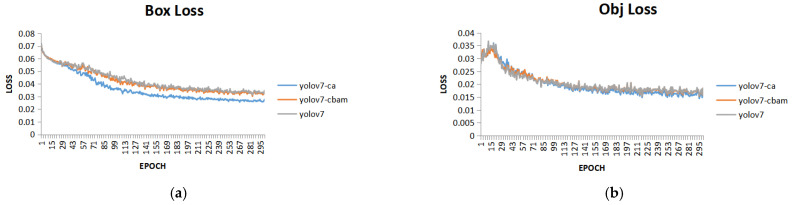
Loss curves for different detector architectures. (**a**) Training curves for different detectors on box loss; (**b**) Training curves for different detectors on Obj loss.

**Table 1 sensors-23-05903-t001:** Judgment indicators and formulas used.

Detection Performance Metrics
$Intersection\;of\;Union\left(IoU\right)=\frac{area\;of\;overlab}{area\;of\;union}$
$Recall\left(R\right)=\frac{TP}{TP+FP}$
$Precision\left(P\right)=\frac{TP}{TP+FP}$
$F_{1}\;score=\frac{2PR}{R+P}$
$Average\;Precision\left(AP\right)=\frac{1}{11}\sum_{R_{i}}PR_{i}$
$Mean\;Average\;Precision\left(mAP\right)=\frac{\sum_{i=1}^{k}AP_{i}}{k}$

**Table 2 sensors-23-05903-t002:** Performance results of different detection models.

Model	Precision	Recall	mAP@0.5	mAP@0.95	F1 Score	Model Size
Yolov7	0.864	0.809	0.873	0.634	0.83	12.3 M
Yolov7t-CBAM	0.896	0.799	0.883	0.625	0.83	12.3 M
Yolov7t-CA	0.924	0.816	0.913	0.638	0.85	12.1 M
Weak-Yolov3 [36]	None	0.766	0.661	None	None	None
Strong-Yolov3 [36]	None	0.887	0.784	None	None	None

**Table 3 sensors-23-05903-t003:** MAE results for different counting models.

Video ID	Manual Counting	Method	Number of Counts	MAE
0	134	SORT	185	0.380
		Cascade-SORT	159	0.186
		Our	148	0.104
		Yolov7	241	0.798
1	92	SORT	113	0.228
		Cascade-SORT	106	0.152
		Our	99	0.076
		Yolov7	157	0.706
2	64	SORT	75	0.171
		Cascade-SORT	68	0.062
		Our	63	0.016
		Yolov7	97	0.426
All	290	SORT	373	0.286
		Cascade-SORT	333	0.148
		Our	310	0.064
		Yolov7	495	0.706

## Data Availability

Publicly available datasets were analyzed in this study. This data can be found here: https://www.ai.rug.nl/~p.pawara/ (accessed on 23 May 2023).

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
