# Peer review of "Fruit Detection and Counting in Apple Orchards Based on Improved Yolov7 and Multi-Object Tracking Methods"

_sensors, 2023, doi:10.3390/s23135903_

Round 1

Reviewer 1 Report

The article presents data of interest to potential readers of the journal. However, I would like to make some comments as recommendations:

Comments

Improve the introduction section, providing the novelty of the work.

The discussion should add the limitations of the study and future directions, as well as discuss the results and explain the practical implications of the work. 

Author Response

Point 1: Improve the introduction section, providing the novelty of the work.

Response 1: Thank you for your guidance. We have added an innovative point about our work at the end of the introduction based on your comments.

Point 2: The discussion should add the limitations of the study and future directions, as well as discuss the results and explain the practical implications of the work.

Response 2: Thank you for your guidance. In the first version, we did not follow the format specified for the paper and therefore did not have a discussion section, but we have now redistributed the paper and added a new discussion section with references to study limitations, findings and how to deploy them.

Reviewer 2 Report

The paper is interesting because it proposes a new variant model that combines the self-attention mechanism of Vision Transform, a non-CNN network architecture, with Yolov7, a state-of-the-art object detection model. This may be important in determining fruit yield and could have implications for decision making in the fruit industry.

The work deserves to be published but major revisions need to be made first:

- Application examples of fruit counting or fruit quality determination using YOLO should be included in the introduction, such as reference: Salvucci, G., Pallottino, F., De Laurentiis, L., Del Frate, F., Manganiello, R., Tocci, F., Vasta, S., Figorilli, S., Bassotti, B., Violino, S., Ortenzi, L., & Antonucci, F. (2022). Fast olive quality assessment through RGB images and advanced convolutional neural network modeling. European Food Research and Technology, 248(5), 1395-1405. Insert more application examples. 

- The 4 section i.e. Experiments is not clear whether it is part of M&M or Results because it reports some things typical of materials and methods and some of results. I suggest you review the division of the different sections well. Also regarding the introduction, you don't need to put the related work in a separate paragraph because it is not a review but an article.

- Improve Table 1 by putting the meaning of the formulas in the notes.

- Improve the formatting of the tables because some are too large and out of text.

- Improve and augment the conclusions by inserting future perspectives.

- Revise the formatting of the bibliography, following the guidelines of the journal.

Author Response

Point 1: Application examples of fruit counting or fruit quality determination using YOLO should be included in the introduction, such as reference: Salvucci, G., Pallottino, F., De Laurentiis, L., Del Frate, F., Manganiello, R., Tocci, F., Vasta, S., Figorilli, S., Bassotti, B., Violino, S., Ortenzi, L., & Antonucci, F. (2022). Fast olive quality assessment through RGB images and advanced convolutional neural network modeling. European Food Research and Technology, 248(5), 1395-1405. Insert more application examples.

Response 1: Thank you for your guidance, we have revised the introduction in line with your comments, adding references including the one you suggested and some other relevant references.

Point 2:  The 4 section i.e. Experiments is not clear whether it is part of M&M or Results because it reports some things typical of materials and methods and some of results. I suggest you review the division of the different sections well. Also regarding the introduction, you don't need to put the related work in a separate paragraph because it is not a review but an article.

Response 2: Thank you for your guidance. Our first version of the paper did not follow the format specified in the thesis and therefore there were many ambiguities. In response to your guidance, we have removed the Related work section and dispersed its contents into other subsections. In addition, we have divided the Experiments section into two sections, Results section and Discussion section, to show the results and discussion of the results respectively, and moved some settings of the experimental methods to the second section.

Point 3:  Improve Table 1 by putting the meaning of the formulas in the notes.

Response 3: Thank you for your guidance, we have adjusted the context of Table1, but we did not find a good way to move the content in it, at a later stage we consider to delete Table1 as a whole and keep only the description part of the indicator.

Point 4:  Improve the formatting of the tables because some are too large and out of text.

Response 4: Thank you for your guidance and we have revised the table formatting in line with the new paper format.

Point 5:  Improve and augment the conclusions by inserting future perspectives.

Response 5: Thank you for your guidance, we have added a new outlook on future work to the conclusion and also added some of the outlooks in the new Discussion section.

Point 6: Revise the formatting of the bibliography, following the guidelines of the journal.

Response 6: Thank you for your guidance, We have adapted the format and overall structure of our bibliography to the official template。

Reviewer 3 Report

The article offers a method for predicting the apple harvest based on the YOLOv7 model. The article is relevant and interesting, has practical significance. To improve the quality of the article, the following changes should be made:

1.) The SORT parameters should be described in more detail: sort_max_age, sort_min_hits, sort_iou_thresh;

2.) The quality of Figure 11 should be improved for a better understanding;

3.) How was the quality of assigning ID (numbers) to apples evaluated, what method was used? It is necessary to describe this study in more detail;

4.) In the conclusions, the results obtained should be compared in more detail with the previously presented studies indicated in the review;

5.) The conclusions should describe in more detail the final best hyperparameters of the model.6.) It would be interesting to describe the prospects for further research, how will the predictive model be built?

Author Response

Point 1: The SORT parameters should be described in more detail: sort_max_age, sort_min_hits, sort_iou_thresh;

Response 1: Thank you for your guidance, we have filled in the description of the parameter settings for SORT in subsection 2.4

Point 2:  The quality of Figure 11 should be improved for a better understanding;

Response 2: Thank you for your guidance, we revisited our Figure 11 and found that it was indeed not clear enough in terms of pixels, we found that this was due to our own lack of HD resolution during the video recording and further loss of detail after the screenshot, after discussion we decided that the graphic was not of great importance and decided to remove it.

Point 3:  How was the quality of assigning ID (numbers) to apples evaluated, what method was used? It is necessary to describe this study in more detail;

Response 3: Thank you for your guidance, we found that at the very beginning we did not consider the issue about ID assignment, so the dataset about agriculture is very sparse, after making the dataset of the detection model, we focused on the performance improvement of the previous detector, for the subsequent counting process, we simply performed manual counting and MAE calculation, without considering the issue about ID assignment, in the discussion section, we add this part to the limitations of this study and enhance our video data to do a more in-depth analysis for ID assignment as well as other metrics in future studies.

Point 4:  In the conclusions, the results obtained should be compared in more detail with the previously presented studies indicated in the review;

Response 4: Thank you for your guidance, in the newly added Discussion section, we compare the results in more depth with previous studies

Point 5:  The conclusions should describe in more detail the final best hyperparameters of the model.

Response 5: Thank you for your guidance, after modifying the template, we have added a new section at the end of discussion about the best results configuration, the specific parameters section is placed by us in subsection 2.5.

Point 6: It would be interesting to describe the prospects for further research, how will the predictive model be built?

Response 6: Thanks to your guidance, we have added an outlook on future research to the Discussion and Conclusion, in addition to a new idea on the construction of predictive models in the Discussion.

Round 2

Reviewer 2 Report

The article can now be accepted for publication, as the authors have followed my earlier suggestions for text improvement. The work now appears to be more complete. Thank you for the work done, very interesting.